# Type 1 Diabetes Induces Hearing Loss: Functional and Histological Findings in An Akita Mouse Model

**DOI:** 10.3390/biomedicines8090343

**Published:** 2020-09-11

**Authors:** Yun Yeong Lee, Yeon Ju Kim, Eun Sol Gil, Hantai Kim, Jeong Hun Jang, Yun-Hoon Choung

**Affiliations:** 1Department of Otolaryngology, Ajou University School of Medicine, Suwon 16499, Korea; seven260@naver.com (Y.Y.L.); yeonju0130@naver.com (Y.J.K.); gilles1996@naver.com (E.S.G.); noto.hantai@gmail.com (H.K.); jhj@ajou.ac.kr (J.H.J.); 2Department of Biomedical Science, Ajou University Graduate School of Medicine, Suwon 16499, Korea

**Keywords:** hearing loss, diabetes, Akita, Na^+^/K^+^-ATPase, apoptosis

## Abstract

The relationship between type 1 diabetes and hearing loss is not well known, although based on many pathological studies, type 2 diabetes induced hearing loss is associated with microcirculation problems in the inner ear. The purpose of this study was to investigate the correlation between type 1 diabetes and hearing loss through hearing function and immunohistochemical analyses using type 1 diabetic Akita or wild-type (WT) mice. The Akita mice had a significant increase in hearing thresholds, blood glucose, and insulin tolerance compared to WT mice. Histological analysis showed that the loss of cells and damage to mitochondria in the spiral ganglion neurons of Akita mice were significantly increased compared to WT. Also, the stria vascularis showed decreased thickness, loss of intermediate cells, and disturbance in blood capillary shape in the Akita mice. Moreover, a reduction in type I, II, and IV fibrocytes and Na^+^/K^+^-ATPase α1 expression in spiral ligament was also observed. Cleaved caspase-3 expression was highly expressed in spiral ganglion neurons. In conclusion, hearing loss in type 1 diabetes is caused not only by ion imbalance and blood flow disorders of cochlear endolymph, but through the degenerative nervous system via apoptosis-mediated cell death.

## 1. Introduction

Hearing loss can be genetic or acquired by exposure to noise, toxic drugs, or systemic diseases such as hypertension, hyperlipidemia, and diabetes [1]. Sensorineural hearing loss (SNHL) occurs mainly because the transmission of sound waves to the brain is disturbed by pathological changes in the inner ear structures such as the cochlea or auditory nerves [2].

Diabetes is classified into types 1 and 2. Type 1 diabetes is caused by gradual loss or dysfunction of insulin-forming β-cells in the pancreas. By contrast, type 2 diabetes is mainly caused by increased insulin resistance and decreased insulin secretion [3]. Diabetes can cause sudden increases in blood viscosity as well as microvascular damage and microcirculation disorders such as thrombosis and embolism [4,5]. Recent analytical studies based on a large population dataset reported that hearing loss in adults with diabetes was twice as great as that in adults without diabetes after adjusting for factors related to hearing loss [6]. A Korean cohort study found a 36% higher risk of incident hearing loss among those with diabetes [7]. The hearing impairment was associated with diabetes complications, such as retinopathy and nephropathy [8].

Another report suggested that type 2 diabetes was also independently associated with a higher risk of multivariate-adjusted incident hearing loss and a longer duration (≥8 years) was associated with a higher risk of moderate or worse hearing loss [9,10]. Moreover, hearing loss was reported to be due to diabetic microangiopathy and macroangiopathy [11]. These reports strongly suggest that diabetes can cause hearing loss.

Histopathological studies of the temporal bone of patients with diabetes reported atrophy or thickening of the stria vascularis (SV) and loss of outer hair cells (OHCs) of the cochlea [9,12]. Also, characteristic changes such as thickening of the cochlea modular vessel walls and microscopic stenosis of the endolymphatic sac and/or basilar membrane vessels have been reported due to diabetes. These reports suggest that microangiopathy is a common feature in the cochlea of diabetic patients [13,14]. Diabetic rodent models also reported a loss of inner hair cells (IHCs) and OHCs, deformation of intermediate and marginal cells of the SV, thickening of basement membranes of capillaries, and degeneration of spiral ganglion neurons (SGNs) [11,15,16,17,18,19].

Sodium-potassium adenosine triphosphatase (Na^+^/K^+^-ATPase) is a membrane-binding enzyme primarily involved in the generation of Na^+^ and K^+^ translocation across plasma membranes and the determination of cellular Na^+^ levels. Na^+^/K^+^-ATPase is ATP-dependent and pumps three Na^+^ ions out of the cell and two K^+^ ions in to create an ionic gradient [20]. This enzyme is essential for maintaining the physiological functions of many cell types by controlling cell volume, intracellular ion balance, and tight junction proteins [21]. Likewise, Na^+^/K^+^-ATPase plays an important role in maintaining the function of the cochlea in the inner ear [22]. Immunohistochemistry studies have shown the presence of this enzyme in the SV, especially marginal cells and the spiral prominence of the cochlear lateral walls (LW) [23]. However, there has been no direct evidence for the involvement of a Na^+^/K^+^-ATPase in diabetic hearing loss.

To date, studies on diabetes and hearing loss have been conducted primarily in type 2 diabetic animal models. Regarding type 1 diabetes, there are reports of morphological findings within the SV of the cochlea such as protrusions of marginal cells, swellings of intermediate cells, and widening of intercellular spaces. However, pathological research on the underlying mechanism of hearing loss is extremely limited. Therefore, the purpose of the study was to investigate the effect of type 1 diabetes on the cochlea and assess morphological changes in association with cell death mechanisms using diabetic C57BL/6-Ins2Akita/J mice.

## 2. Material and Methods

### 2.1. Animals

Age-matched male C57BL/6J WT and Akita mice (7 weeks) were obtained from Dae Han Bio Link (Chungbuk, Korea). All animal procedures were approved by the Institutional Animal Care and Use Committee guidelines of the Ajou University Graduate School of Medicine (Permit Number: 2017-0004, approval date: 02-10-2017). The animals were maintained under standard animal housing conditions.

### 2.2. Analysis of the Auditory Brainstem Response (ABR)

The ABR threshold was defined as the lowest stimulus level at which a clear waveform was visible in the evoked trace using the TDT II System (Tucker-Davis Technologies, Alachua, FL, USA) and BioSig software (MathWorks, Natick, MA, USA). The ABR was tested as described previously [24]. Briefly, mice were anesthetized by intraperitoneal (i.p.) administration of ketamine (125 mg/kg) and xylazine (2.5 mg/kg). For hearing threshold evaluation, needle electrodes were inserted subcutaneously at the vertex (active), under the pinna of the left ear (reference), and the right ear (ground). ABRs were measured at frequencies of 8, 16, and 32 kHz with tone-burst stimuli reducing levels in the range of 10–80 dB and 10-dB intervals to determine the lowest intensity level. Each measurement point was recorded 1000 times and averaged. The core temperature was maintained at 37 °C using a heating pad.

### 2.3. Metabolic Studies

The level of glucose in blood obtained from the tail vein was measured using an Accu-Chek Active Glucose Meter (Roche Diagnostics, Mannheim, Germany) at baseline and 1-month intervals from 3 weeks until the end of the study. An intraperitoneal insulin tolerance test (IPITT) was used to evaluate eight mice per group at 1 to 3 months. Mice received intraperitoneal injection administration of 0.75 U/kg regular human insulin (Novo Nordisk, Clayton, NC, USA) between 2:00 p.m. and 5:00 p.m. Blood samples were obtained from the tail vein and glucose levels were measured at 0, 15, 30, 60, 90, and 120 min by the sensitive strip method using a blood glucose monitoring system (i-SENS, Seoul, Korea). The integrated area under the curve (AUC) was calculated.

### 2.4. Haematoxylin and Eosin (H&E) and Immunohistochemistry Staining

The cochlea was dissected, fixed with 4% paraformaldehyde, and decalcified in Calci-Clear Rapid Decalcifying Solution (National Diagnostics, Atlanta, GA, USA) for 4 days. Decalcified cochleae were embedded in paraffin. For immunofluorescent histological analysis, paraffin-embedded sections were cut to 6 μm thick. Cochlear and liver sections were first dewaxed in xylene and rehydrated through a series of graded ethanol washes (100, 90, 80, and 70%), then subjected to histological analysis using H&E. For antigen retrieval immunohistochemistry experiments, the slides were placed in 10 mM sodium citrate buffer (pH 6.0) and heated to near boiling (95–98 °C) in a water bath for 30 min followed by cooling for 30 min at room temperature. Then, cochlear sections were blocked with endogenous peroxidase with 3% hydrogen peroxide (Sigma, St. Louis, MO, USA) for 15 min. Cochlear sections were then incubated for 1 h at room temperature in blocking/permeabilization solution containing 3% bovine serum albumin (GenDEPOT, Barker, TX, USA) and Triton X-100 (0.05%) in 0.1 M phosphate-buffered saline (PBS). Next, the sections were incubated with primary antibodies overnight at 4 °C. After three washes with 0.1 M PBS, sections were incubated for 1 h at room temperature with horseradish peroxidase-conjugated secondary antibodies. Sections were counterstained with hematoxylin for each analysis. Negative controls for the immunohistochemical procedures included substitution of the primary antibody with nonimmune sera. Samples were dehydrated and cleared in xylene, then coverslipped using a permanent mounting medium (Fisher Scientific, Pittsburgh, PA, USA). Sections were observed and images were captured using bright-field microscopy (Olympus BX-60, Tokyo, Japan).

### 2.5. Electron Microscopy

Whole cochlear samples were post-fixed in 1% osmium tetroxide, dehydrated in 70–100% ethanol, incubated in propylene oxide, and embedded in Embed 812 resin (Electronic Microscopic Science, Hatfield, PA, USA). Cochlea histology was observed under an EM902A microscope (Carl Zeiss MicroImaging, Oberkochen, Germany) at the specified magnification.

### 2.6. SGN and Spiral Ligament (SL) Counts

Morphometric assessments of the SGNs and SLs were performed for the middle or apical to basal cochlear turn on H&E-stained sections. The cochlear specimens were observed and photographed using bright-field microscopy (Olympus BX-60) and digital images were saved. The area of the cochlear turns was quantified using Image J software (version 1.52h; National Institutes of Health Bethesda, MD, USA). The SGN density was determined as the number of cell nuclei per 10,000 μm^2^ Rosenthal canal on the middle turn. The SL was divided into four parts based on cell types and density was determined as the number of cell nuclei per each part in 10,000 μm^2^.

### 2.7. Assessment of SV Thickness

The thickness of the SV was assessed with H&E-stained cochlear sections. At the apical to basal cochlear turn in each cochlea section, five images containing the entire thickness of the SV (from the endolymphatic surface to the interface with the SL) were taken from the central portion of the strial width (along the axis of Reissner’s membrane to the spiral prominence). Using Image J, measurements of the thickness of the SV were obtained from each image.

### 2.8. CD31, Na^+^/K^+^-ATPase α1, Cleaved Caspase-3, and Intermediate Cell Intensity Quantitation

Eosin-positive intermediate cell intensity was quantified based on pixel intensity/area using Image J. The sections at the mid-cochlear turns were selected as regions of interest (ROIs) for immunolabeling. Using the freehand selection tool, we selected the 3,3′-diaminobenzidine-stained ROIs and calculated the pixel intensity/area. For intensity measurements, the mean gray value was determined by converting the RGB pixels in the image to grayscale/brightness values. The mean gray value represents the sum of the gray values of all pixels in the selection divided by the total number of pixels. The lower the pixel value, the higher the intensity. The mean gray values and areas of the ROIs were averaged for three independent sections and presented as relative intensity compared to the control group.

### 2.9. Statistical Analysis

All data are presented as means ± standard error of the mean (SEM) of at least three independent experiments. The significance of differences in quantitative data was analyzed by Student’s *t*-test (two groups). One-way analysis of variance (ANOVA) followed by Tukey’s honest significant difference (HSD) post hoc test and two-way ANOVA with Šídák’s multiple comparisons tests were conducted using the SPSS software version 22.0 (SPSS Inc., Chicago, IL, USA). A *p*-value < 0.05 was considered to indicate significance.

### 2.10. Antibodies

Antibodies against anti-CD31 (ab124432) and Na^+^/K^+^-ATPase α1 (ab7671) were purchased from (Abcam, Cambridge, UK); cleaved caspase-3 (9961) was purchased from (Cell Signaling Technology, MA, USA).

## 3. Results

### 3.1. Akita Mice with Type 1 Diabetes Exhibit Hearing Loss

Akita mouse is an animal model of type 1 diabetes caused by pancreatic β-cell failure [25]. We first evaluated fasting glucose levels in wild-type (WT) and Akita mice (Figure 1A). The Akita group had significantly impaired glucose levels (145 ± 12.3 vs. 516 ± 53.2) and insulin sensitivity compared to the WT group at 0, 15, 30, 60, and 120 min after subcutaneous injection of insulin (AUC: IPITT, 477 ± 33 vs. 1632 ± 66) (Figure 1B,C). At 3 months, the Akita group showed significantly increased ABR thresholds at 16 and 32 kHz compared to the WT group (16 kHz, 5.6 dB ± 1.2; 32 kHz, 5.9 dB ± 1.1 shift) (Figure 1D). This result indicates that type 1 diabetes causes hearing impairment.

### 3.2. Type 1 Diabetes Induces Damage in the SGNs but Not the Organ of Corti (OC)

Degeneration of the OC and SGNs are an important component of SNHL. The histopathological changes were focused on the OC and the SGNs in the cochleae of the Akita and WT groups (Figure 2A,B). The most significant finding in the cochleae of the Akita group was the decrease of the number of SGNs (Figure 2F, red arrow; Figure 2I). At the ultramicroscopic level, SGNs in Akita mice showed markedly abnormal mitochondria compared to the WT group, resulting in disruption of normal structures of the SGNs (Figure 2G,H,J). Similar results were observed with H&E staining (Figure 2D,F). By contrast, the IHCs, OHCs, and supporting cells (SCs) showed no histopathological changes in WT or Akita mice (Figure 2C,E).

### 3.3. Type 1 Diabetes Induces Degeneration of the SV

The cochlear LW, including the SL and SV, plays an integral role in the creation of the endolymphatic potential. In the Akita group, the thickness of the SV was significantly decreased in the middle to basal turns of the cochlea compared to the WT group (Figure 3A,B). The SV comprises three cell types: marginal, intermediate, and basal. Ultrastructural observations of the Akita group revealed several cavities in marginal, intermediate, and basal cells (Figure 3C). In particular, the intermediate cell density was significantly decreased (Figure 3D); this decrease was quantified in Figure 3F. The Akita group also showed abnormal shapes in both the blood-labyrinth barrier (BLL, red dotted line) and capillary formations (yellow dotted line). These changes were not detected in the WT group (Figure 3C). The vessel walls of the SV in the cochlea were stained positively for cluster of differentiation 31 (CD31) [26]. However, CD31 was significantly decreased in Akita mice compared to WT mice (Figure 3E,G). Thus, the decrease in CD31 expression at the SV may be correlated with the altered capillary shapes in the Akita group.

### 3.4. Type 1 Diabetes Induces a Decrease in Type I, II, and IV Fibrocytes in the Spiral Ligament

The fibrocytes of the SL are divided into four cell types based on histological characteristics, immunostaining patterns, and general location [27,28]. We analyzed the histology of the SL based on a schematic diagram of the four fibrocyte classes in the SL (Figure 4A). Interestingly, severe shrinkage and a significant decrease in the cell numbers of type I fibrocytes was observed from the apical to basal sections in Akita mice compared to WT (Figure 4B). Also, type II and IV fibrocytes showed significantly decreased cell numbers in the middle and basal sections, but not the apical turn, in the Akita group compared to the WT group. These changes were quantitated and are shown in Figure 4C.

### 3.5. Type 1 Diabetes Induces a Decrease in Na^+^/K^+^-Atpase A1 Expression in the SL and SGNs

Na^+^/K^+^-ATPase is involved in specialized functions in the ear [22] and is detectable in the SV and cochlear LW structures [23]. Multiple isoforms of three subunits (α, β, and γ) constitute the Na^+^/K^+^-ATPase oligomer. Of these subunits, the α subunit has binding sites for ATP and cations (Na^+^ and K^+^). Thus, we determined the expression levels of Na^+^/K^+^-ATPase α1 at the cochlea in WT and Akita mice. IHC data revealed that Na^+^/K^+^-ATPase α1 was expressed predominantly in type I, II, and IV fibrocytes, the entire SV, and the OC in the apical turn in the Akita mice. However, its expression was detected in type II and IV fibrocytes and the SV in the middle to basal turns in the WT mice (Figure 5A, upper row). Moreover, in the Akita group, the expression in type II and IV fibrocytes and the SV in the middle to basal turn was decreased in comparison to the WT group (Figure 5A, bottom row). These changes were quantitated and are shown in Figure 5B. Furthermore, the expression of Na^+^/K^+^-ATPase α1 was marked in the SGNs of the WT group but was significantly decreased in the Akita group (Figure 5C,D).

### 3.6. Activation of Caspase-3 is Associated with Degeneration of the SV and SL in the Type 1 Diabetes Cochlea

Apoptosis occurs through the sequential actions of caspases, such as caspase-3, which is initiated by their associated extrinsic and intrinsic pathways [29]. The caspase-dependent cell death pathway has been widely implicated in the programmed cell death of IHCs and OHCs [30,31]. Also, a previous study found that an apoptosis index was increased in fibroblast cells of the cochlear LW in gentamicin (ototoxic drug)-induced hearing-impaired rat cochleae [32]. Our data showed higher expression of cleaved caspase-3 in the SV and type I, II, and IV fibrocytes of Akita mice compared to the WT group (Figure 6A,B). Cleaved caspase-3 was strongly expressed in the basal turn of the OC and nerve fibers. The changes were observed and quantified in the SGNs (Figure 6C,D). This result is similar to the reduction in SV thickness, the loss of type I, II, and IV fibrocytes and SGNs, and the reduction in Na^+^/K^+^-ATPase α1.

## 4. Discussion

Diabetes has been reported to be both clinically [33,34,35] and pre-clinically [36,37] correlated with hearing loss. The results of the present study showed that hearing loss in type 1 diabetic Akita mice was increased compared to WT (Figure 1). Hearing loss was most pronounced at the 16 and 32 kHz frequencies. This finding is in agreement with previous studies that showed that hearing loss is induced by spontaneous diabetes mellitus along with glucose intolerance in WBN/Kob rats [15,38] as well as a report based on a meta-analysis of currently available published data that diabetes mellitus is associated with an increased risk for developing hearing loss [39].

Sensory hair cells located in the OC are responsible for sound reception and signal transmission to the brain through SGNs. Many studies have shown structural changes in the inner ear in hearing loss, most of which focused on damage to the auditory cells and SGNs. [40,41,42]. Much less is known about how damage to the SV and SL contributes to hearing loss. A loss of SGNs and degeneration of OHCs were observed in the middle and basal turns of the cochlea in ob/ob mouse models [43]. However, loss of SGNs without the associated loss of hair cells is common among mammals during aging [44,45]. Moreover, diabetic mice exhibited significant loss of SGNs in the cochlea after noise exposure, while IHCs and OHCs were preserved. [46]. In our data, the WT and Akita groups had no change in IHCs and OHCs (Figure 2C,E). However, the loss of SGNs, including mitochondrial damage, observed in the Akita group (Figure 2A–C) was correlated with caspase 3 expression (Figure 6C). This means that SGNs are reduced through the mitochondria-mediated intrinsic apoptosis pathway.

Previous studies of cochlear structures associated with type 2 diabetes have shown that microangiopathy can affect inner vascularization, and thickening of the capillary walls can interfere with nutrient transport by narrowing blood vessels and reducing blood flow [11,47]. In our data, the Akita group showed significantly decreased thickness of the SV in the middle to basal turns, as well as abnormalities of capillaries and blood labyrinth layers in the SV (Figure 3A–E). In addition, decreases in both intermediate and vessel endothelial cells were observed by H&E and immunostaining with CD31 (Figure 3D–G). This seems to cause stenosis of blood vessels due to cell loss, unlike the results reported in models of type 2 diabetes. Also, when compared with cleaved caspase 3 expression (Figure 6A,B) and SV histological analysis, the ABR threshold results of the basal (32 kHz) to middle (16 kHz) turns (Figure 1D) were well correlated. As a result, type 1 diabetes showed microangiopathy, implying that it is an important factor in hearing loss as a result of limiting blood flow. The microangiopathy and subsequent degeneration of the lateral cochlear walls detected in the present study have also been detected in humans with type 1 diabetes [9]. However, human data include some different findings of OHC degeneration and sustained number of SGNs. These differences may be a result of the observation time, glucose levels, or species differences. Nevertheless, the similar clinical and histological findings of hearing loss make Akita mice a useful animal model for type 1 diabetes

Sensory hair cells are surrounded in potassium-rich endolymph, the composition of which is essential for hair cell depolarization and signal transduction. Genetic mutations in potassium channels are causes of deafness [48,49]. SV cells and fibrocytes of the SL contribute to potassium recycling and endolymph homeostasis [50]. Fibrocytes of the SL are divided into four cell types based on histological characteristics [27,28]. The structural characteristics of these cells are illustrated in Figure 4A. Type I fibrocytes are located adjacent to the SV and are in gap junction continuity with the basal cells of the SV [28]. Type II fibrocytes are localized in the region near the spiral prominence epithelium. Type I and II fibrocytes potentially play a role in potassium recycling [27,28,51]. Type III fibrocytes line the bony otic capsule and are oriented perpendicular to the plane of the section shown in Figure 4. Type IV fibrocytes are spindle-shaped and organelle-poor and are the only fibrocyte type within a triangular space inferior to the basilar crest. In our data, in the Akita group, type I, II, and IV fibrocytes were significantly decreased at the middle to basal turns and type I fibrocytes were decreased at the apical turn of the SL (Figure 4B,C). When compared to cleaved caspase-3 expression, cell reduction in the SV and SL through apoptosis appears to cause an ion imbalance (Figure 6A,B).

We examined the expression of Na^+^/K^+^-ATPase α1 in the LW structures of WT and Akita mice. Histological analysis by immunostaining revealed that the enzyme was located predominantly in type II and IV fibrocytes, the SV, and SGNs in the WT group. However, the expression level was significantly decreased in the middle to basal turns in the Akita group (Figure 5A,B). Furthermore, cleaved caspase-3 was expressed mainly in type I, II, and IV fibrocytes, the SV, and SGNs in the middle to basal turns with partial expression in the apical turn in the Akita group (Figure 6A,B). Meanwhile, no significant change in Na^+^/K^+^-ATPase α1 expression was observed in the apical turn. Although cleaved caspase-3 expression was observed in the apical turn, the absence of a change in Na^+^/K^+^-ATPase α1 expression was probably due to the higher protection mechanism in the apical turn compared to the middle and basal turns.

## 5. Conclusions

We demonstrated that loss of SV, SL, SGNs, and Na^+^/K^+^-ATPase α1 was due mainly to apoptosis. Further research into the mechanisms that induce cell death may provide clues for preventing and treating diabetes-mediated hearing loss. Type 1 diabetic Akita mice exhibited hearing loss associated with the disruption of ion trafficking and blood circulation, involving a decrease in SV, SL, and Na^+^/K^+^-ATPase α1 in the LW structures. Moreover, limited sound transmission to the brain was caused by mitochondrial abnormalities and cell death in the SGNs.

## Figures and Tables

**Figure 1 biomedicines-08-00343-f001:**
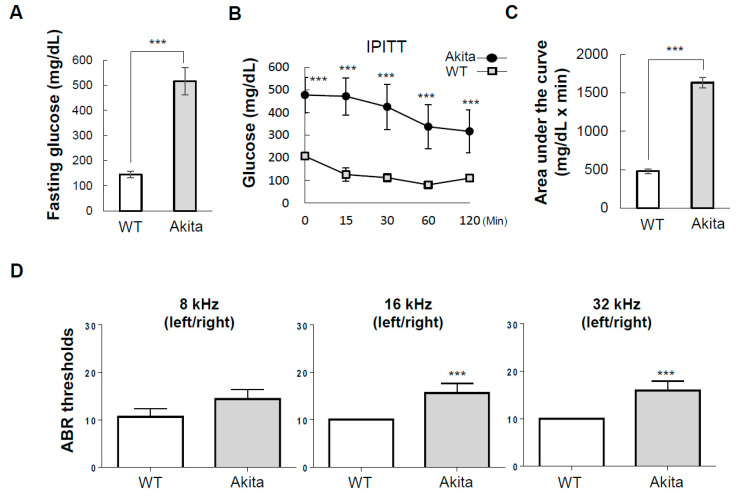
Type 1 diabetic Akita mice exhibited hearing impairment. (**A**) Akita mice had higher blood glucose and (**B**) insulin tolerance compared to WT mice. (**C**) Area under the curve (AUC) of the results of intraperitoneal insulin tolerance test (IPITT). (**D**) Auditory brainstem response (ABR) thresholds at 16 and 32 kHz were significantly elevated in the Akita group (*n* = 8) compared to the wild type (WT) group (*n* = 8) at 3 months (16 kHz, 10 dB ± 1.2; 32 kHz, 9 dB ± 1.1 shift). Results are means ± SEM. *** *p* < 0.001 by Student’s *t*-test (**A**,**C**) or two-way ANOVA followed by Šídák’s multiple comparisons test (**B**,**D**).

**Figure 2 biomedicines-08-00343-f002:**
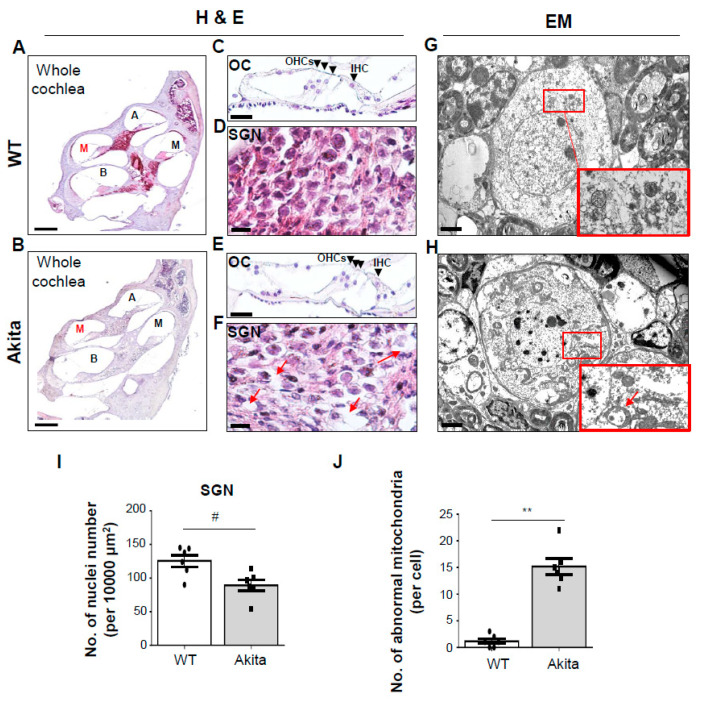
Histopathological analysis of the organ of Corti (OC) and spiral ganglion neurons (SGNs). (**A**,**B**) H&E staining of WT and Akita cochleae. Whole cochleae (scale bar, 250 μm), (**C**,**E**) OCs (scale bar, 30 μm), and (**D**,**F**) SGNs in the middle turn of the cochlea (scale bar, 10 μm). The black arrowheads indicate inner hair cells (IHCs) and outer hair cells (OHCs); the loss of SGNs is indicated by a red arrow. (**G**,**H**) Transmission electron micrograph (TEM) of SGNs in the middle turns of WT and Akita cochleae. Red boxes are zoom-ins showing mitochondria morphology. Abnormal mitochondria shapes (red arrows) can frequently be seen in the Akita group compared to WT. Scale bar, 2 μm. (**I**) Quantification of the number of SGNs in the middle turns of the cochlea. The average number of SGNs per 10,000 μm^2^ in the middle turns of cochlea tissue from WT and Akita mice. Individual squares or circles represent individual mice (*n* = 6 for each group). (**J**) The numbers of abnormal mitochondria between WT and Akita mice were quantified using Image J (*n* = 6 images/group). Results are means ± SEM. # *p* < 0.05 and ** *p* < 0.01 by Student’s *t*-test.

**Figure 3 biomedicines-08-00343-f003:**
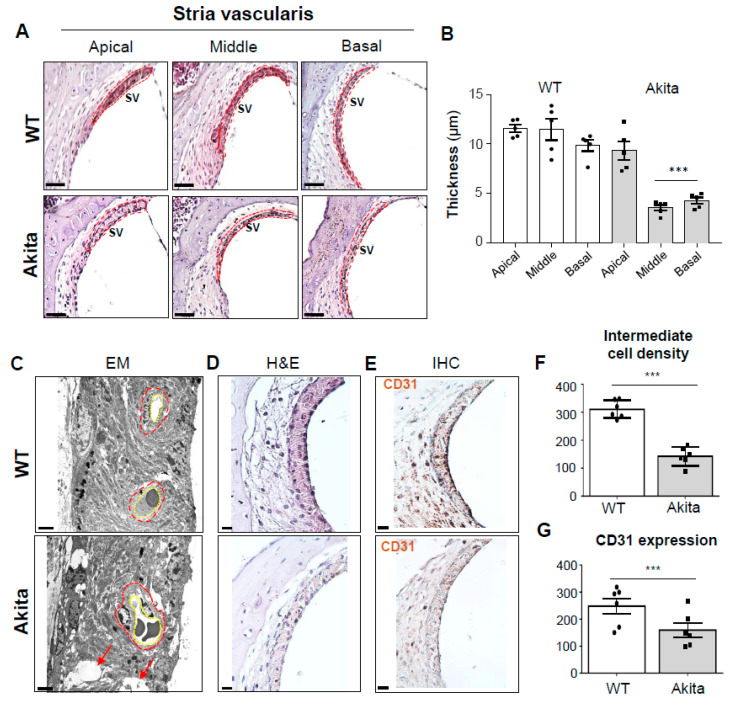
Comparative study of the stria vascularis (SV) in WT and Akita mice. (**A**) H&E-stained sections from the SV (red dotted line) in the apical, middle, and basal turns of the cochlea. Scale bars, 50 μm. (**B**) Graph representing the quantification of SV thickness in the apical, middle, and basal turns of WT and Akita mice. Individual circles represent individual mice (*n* = 5, for each group). (**C**) TEMs of WT and Akita (at the mid-turn) mice showing the blood-labyrinth barrier (BLL) (red dotted line), capillary pores (yellow dotted line), and intermediate cells with cavities (red arrow) throughout the SVs. Scale bars, 2 μm. (**D**) H&E staining of intermediate cell density and (**E**) CD31 immunostaining of the vascular endothelial cells in the SV of the mid-turns in WT and Akita mice. Scale bars, 20 μm. (**F**) Intermediate cell density and (**G**) chromogenic intensity of CD31 quantified by image J analysis are shown for both WT (*n* = 6) and Akita (*n* = 6) mice. Results are means ± SEM. *** *p* < 0.001 by Student’s *t*-test.

**Figure 4 biomedicines-08-00343-f004:**
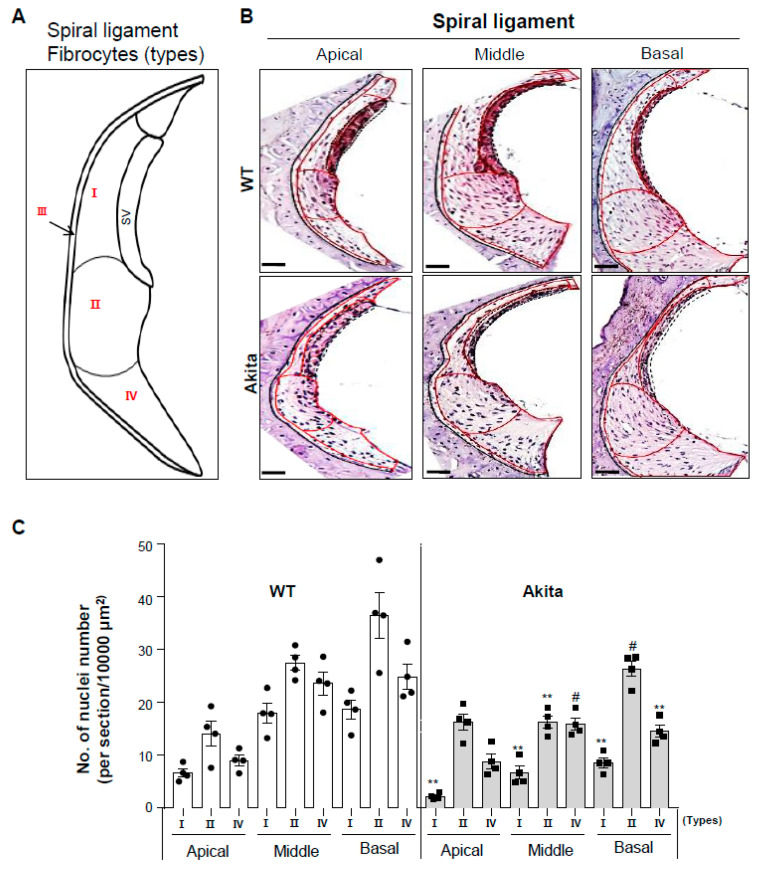
Comparative study of the spiral ligament (SL) in cochleae of WT and Akita mice. (**A**) Diagram of various types (I–IV) of fibrocytes in the lateral wall (LW) structures. (**B**) H&E staining of WT (upper) and Akita mice (lower) showing the four types of SL separated by a red line. Scale bar, 50 μm. (**C**) The Akita mice had an abnormal morphology with a loss of nuclei in types I, II, and IV. The graph shows the quantification of hematoxylin-stained nuclei in SL type I, II, and IV fibrocytes in the apical, middle, and basal turns of Akita mice compared to WT. Individual circles represent individual mice (*n* = 4 for each group). Scale bar, 50 μm. Results are means ± SEM. # *p* < 0.05, ** *p* < 0.01 by one-way ANOVA followed by Tukey’s HSD post hoc test.

**Figure 5 biomedicines-08-00343-f005:**
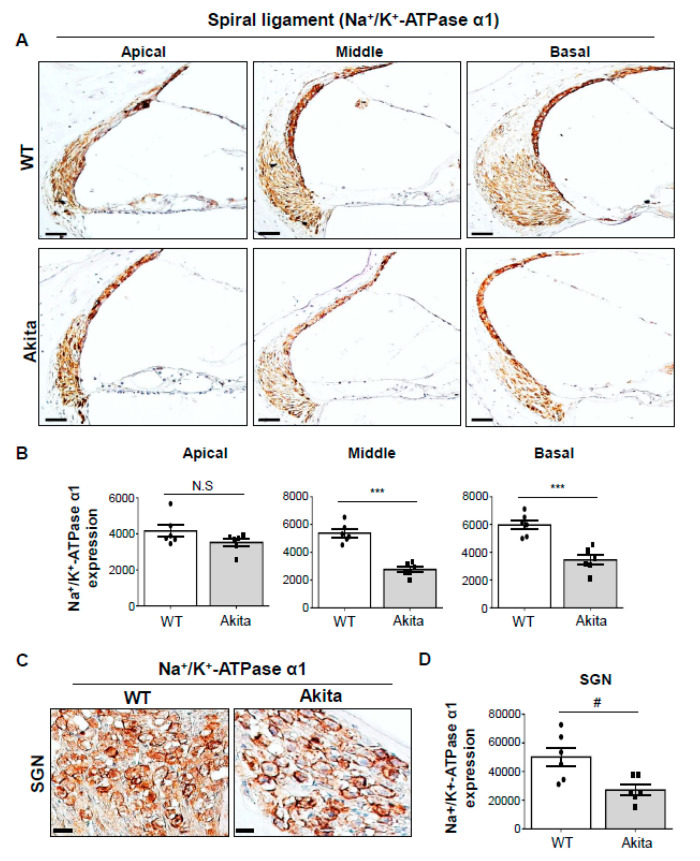
Quantitative analysis of Na^+^/K^+^-ATPase α1 immunostaining in the cochleae of WT and Akita mice. (**A**) Immunohistochemistry analysis of Na^+^/K^+^-ATPase α1 from the apical, middle, and basal turns of WT and Akita mice. Scale bar, 50 μm. (**B**) The chromogenic intensity of Na^+^/K^+^-ATPase α1 quantified by Image J analysis is shown for both WT (*n* = 6) and Akita mice (*n* = 6). (**C**) Immunohistochemistry analysis of Na^+^/K^+^-ATPase α1 from the middle turns of the modiolus of SGNs in WT and Akita mice. Scale bar, 10 μm. (**D**) The chromogenic intensity of Na^+^/K^+^-ATPase α1 quantified by Image J analysis is shown for both WT (*n* = 6) and Akita mice (*n* = 6). Results are means ± SEM. # *p* < 0.05, *** *p* < 0.001 by Student’s *t*-test. N.S = not statistically significant.

**Figure 6 biomedicines-08-00343-f006:**
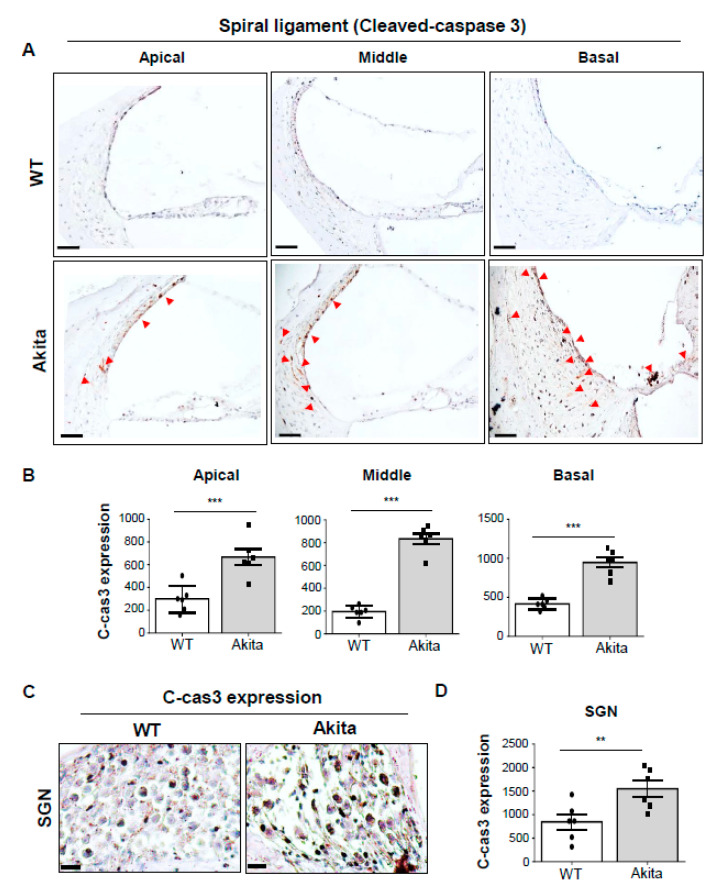
Quantitative analysis of activated caspase-3 immunostaining in the cochleae of WT and Akita mice. (**A**) Immunohistochemistry analysis of cleaved caspase-3 (c-cas3) from the apical, middle, and basal turns of WT and Akita mice. The expression of c-cas3 is marked by red arrowheads in the Akita group. Scale bar, 50 μm. (**B**) The chromogenic intensity of c-cas3 quantified by Image J analysis, shown for both WT (*n* = 6) and Akita mice (*n* = 6). (**C**) Immunohistochemistry analysis of c-cas3 from the middle turns of the modiolus of SGNs in WT and Akita mice. Scale bar, 10 μm. (**D**) The chromogenic intensity of c-cas3 quantified by Image J analysis, shown for both WT (*n* = 6) and Akita mice (*n* = 6). Results are means ± SEM. ** *p* < 0.01 or *** *p* < 0.001 by Student’s *t*-test.

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
