# Peer review of "Type 1 Diabetes Induces Hearing Loss: Functional and Histological Findings in An Akita Mouse Model"

_biomedicines, 2020, doi:10.3390/biomedicines8090343_

Round 1

Reviewer 1 Report

Interesting and well written paper.

I have two questions/comments:

1) Why was Akita mouse model chosen ?

2) In the Introduction the authors could provide some data on hearing problems in human diabetic patiens. How often do you find them, what are the risk factors known sofar (duration of diabetes, presence of nephropathy, retinopathy).

Author Response

Response to Reviewer

Reviewer 1:

Comments and Suggestions for Authors

Interesting and well written paper.

I have two questions/comments:

1) Why was Akita mouse model chosen?

Answers

Currently, there are four main types of type 1 diabetic mouse models: 1) chemically induced type 1 diabetes; 2) spontaneous autoimmune models of type 1 diabetes; 3) genetically induced insulin-dependent diabetes; and 4) virus-induced models of diabetes. Chemically induced models are usually induced with streptozotocin (STZ), an alkylating agent that targets insulin-producing beta cells in the pancreas. However, STZ can cause acute kidney damage in animals via non-specific cytotoxicity, which makes it difficult to differentiate between the direct toxic effects of STZ and lesions caused by hyperglycemia (Ref. 1). Therefore, in this study, a genetically induced insulin-dependent diabetes Akita mouse was used to obtain more reliable data on type 1 diabetes-induced hearing loss. Other models were excluded because they were not suitable for this study.

Ref.1.) M. D. Breyer, E. Bottinger, F. C. Brosius III et al., “Mouse models of diabetic nephropathy,” Journal of the American Society of Nephrology, vol. 16, no. 1, pp. 27–45, 2005.

2) In the Introduction the authors could provide some data on hearing problems in human diabetic patients. How often do you find them, what are the risk factors known so far (duration of diabetes, presence of nephropathy, retinopathy).

Answer

Thank you for your comments. As you recommend, we revised the manuscript (lines 36-45, pages 1-2).

  • lines 36-45, page 1-2

Recent analytical studies based on a large population dataset reported that hearing loss in adults with diabetes was twice as great as that in adults without diabetes after adjusting for factors related to hearing loss [6]. A Korean cohort study found a 36% higher risk of incident hearing loss among those with diabetes [7]. The hearing impairment was associated with diabetes complications, such as retinopathy and nephropathy [8].

Another report suggested that type 2 diabetes was also independently associated with a higher risk of multivariate-adjusted incident hearing loss and a longer duration (≥ 8 years) was associated with a higher risk of moderate or worse hearing loss [9, 10]. Moreover, hearing loss was reported to be due to diabetic microangiopathy and macroangiopathy [11]. These reports strongly suggest that diabetes can cause hearing loss.

Reviewer 2 Report

Reviewed is a well-designed and executed project looking into the function and histology of a mouse model of Insulin dependent diabetes.

One comment: the authors report a reduction in the number of spiral ganglion neurons. It is worth adding in the discussion that this observation is debatable in humans, as evident, for example, in the conflicting human data provided in reference # 8. The distinction has an important clinical implication.

Author Response

Response to Reviewer

Reviewer 2:

Comments and Suggestions for Authors

Reviewed is a well-designed and executed project looking into the function and histology of a mouse model of Insulin dependent diabetes.

One comment: the authors report a reduction in the number of spiral ganglion neurons. It is worth adding in the discussion that this observation is debatable in humans, as evident, for example, in the conflicting human data provided in reference # 8. The distinction has an important clinical implication.

Answer

Thank you very much for your kind comment. Following your suggestion, we added the explanation into the discussion part, as follows (line 306-311, page 11).

“The microangiopathy and subsequent degeneration of the lateral cochlear walls detected in the present study have also been detected in humans with type 1 diabetes [9]. However, human data includes some different findings of OHC degeneration and sustained number of SGNs. These differences may be a result of the observation time, glucose levels, or species differences. Nevertheless, the similar clinical and histological findings of hearing loss make Akita mice a useful animal model for type 1 diabetes.
